# Trivializations for Gradient-Based Optimization on Manifolds

**Mario Lezcano-Casado**
Department of Mathematics
University of Oxford
Oxford,
`mario.lezcanocasado@maths.ox.ac.uk`

## Abstract

We introduce a framework to study the transformation of problems with manifold constraints into unconstrained problems through parametrizations in terms of a Euclidean space. We call these parametrizations *trivializations*. We prove conditions under which a trivialization is sound in the context of gradient-based optimization and we show how two large families of trivializations have overall favorable properties, but also suffer from a performance issue. We then introduce *dynamic trivializations*, which solve this problem, and we show how these form a family of optimization methods that lie between trivializations and Riemannian gradient descent, and combine the benefits of both of them. We then show how to implement these two families of trivializations in practice for different matrix manifolds. To this end, we prove a formula for the gradient of the exponential of matrices, which can be of practical interest on its own. Finally, we show how dynamic trivializations improve the performance of existing methods on standard tasks designed to test long-term memory within neural networks.[1]

## 1   Introduction

Constrained optimization allows to put restrictions on the family of objects being optimized. When the restrictions are simple, for example, having a vector with entries in $[0, 1]$ or $[-1, 1]$, simple element-wise parametrizations using sigmoid functions or $\tanh$ allow the design of powerful models such as LSTM [Hochreiter and Schmidhuber, 1997] and GRU [Cho et al., 2014] through the method of *gating*. This kind of vector-regularization is now standard, and most of the advanced neural network architectures use it as a basic building block [Bahdanau et al., 2014]. Constraints on matrices, on the other hand, are much more challenging.

Most of the interesting sets of matrices turn out to have a manifold structure. Optimization on manifolds is both theoretically and practically challenging due to the inherent complexity of the objects involved. Even then, optimization on matrix manifolds has proven to be rather useful in many different subfields of machine learning and neural networks (NN). Examples of interesting matrix manifolds in the context of gradient-based optimization are the set of positive definite matrices in Bayesian statistics [Rasmussen and Williams, 2005], orthogonal matrices within RNNs [Arjovsky et al., 2016, Helfrich et al., 2018, Lezcano-Casado and Martínez-Rubio, 2019], NNs with structured linear layers via the QR or the SVD decomposition [Berg et al., 2018, Zhang et al., 2018, Kingma and Dhariwal, 2018], or invertible matrices in normalizing flows [Berg et al., 2018] and VAEs [Tomczak and Welling, 2016].

In this paper we aim to provide a theoretically sound but also efficiently implementable framework to perform optimization on these and other matrix manifolds in the context of gradient-based optimization.

**Outline of the paper and summary of the main contributions**
In this paper, we study parametrizations of the form $\phi\colon \mathbb{R}^n \to \mathcal{M}$.

We consider the transformation of a constrained optimization problem into an unconstrained one.

$$\text{Initial problem:} \min_{x \in \mathcal{M}} f(x) \qquad \text{Unconstrained problem:} \min_{y \in \mathbb{R}^n} f(\phi(y)).$$

We call this process *trivialization* and we say that $\phi$ is a trivialization map. In Section 4, we show that whenever $\phi$ is regular enough— a diffeomorphism—these parametrizations act as a change of metric on $\mathcal{M}$, and thus, applying gradient descent to this new problem is equivalent to performing RGD on the original problem with this new metric, for which standard convergence results hold.

After this, we look at two large families of parametrizations, the Riemannian exponential, and the Lie exponential. We analyze these from the point of view of the framework presented before, and we point out a problem that they present: they may create saddle points or local minima when near certain region in the manifold.

In Section 5, we introduce *dynamic trivializations*. They can be described as follows:

**Main idea:** Lift the function $f$ to the current tangent space $T_{x_i}\mathcal{M}$ using a map $r_{x_i}\colon T_{x_i}\mathcal{M} \to \mathcal{M}$ by considering the trivialization $f \circ r_{x_i}$ (think $r_{x_i} = \exp_{x_i}$, or, for efficiency, any retraction). Optimize $f \circ r_{x_i}$ on $T_{x_i}\mathcal{M}$ for a while using any standard optimization methods like ADAM, RMSPROP, or ADAGRAD, since $T_{x_i}\mathcal{M}$ is a linear space. When we are at a point $y_k \in T_{x_i}\mathcal{M}$ on which $r_{x_i}$ might create saddle-points or local minima, then we consider the current point in the manifold $x_{i+1} := r_{x_i}(y_k)$ and we start optimizing the function $f \circ r_{x_{i+1}}$, *i.e.*, lift the problem to $T_{x_{i+1}}\mathcal{M}$.

This family of methods has Riemannian gradient descent and classic trivializations as limit cases, and in particular, they combine the strengths of the two. Furthermore, we show that these methods give a natural generalization of Euclidean optimizers to manifolds.

In Section 6 we show how to compute the gradients associated to the Lie exponential and some cases of the Riemannian exponential for matrix manifolds. To this end, we compute a formula that allows for the approximation of the gradient of the exponential of matrices to machine-precision. We also show some examples of for how to use this theory to perform optimization on some matrix manifolds. In Appendix E we compile an extended list of examples that we hope might be helpful to the reader.

Finally, in Section 7 we show how dynamic trivializations improve previously developed optimization techniques in the context of optimization with orthogonal constraints.

## 2   Related Work

**Optimization on manifolds.**   Most of the results on optimization on manifolds have found analogues in the Riemannian setting [Udriste, 1994, Absil et al., 2009]. Algorithms like conjugate gradient descent or the Newton method were first devised for specific families of manifolds [Smith, 1993, Edelman et al., 1998], and then they were derived for general Riemannian manifolds [Bonnabel, 2013, Sato and Iwai, 2015, Boumal et al., 2016].

Optimization methods on manifolds can be classified in two families: Those that follow geodesics, and those that follow retractions—*i.e.*, first order approximations to geodesics. In the first family, convergence rates have been proven for most first order methods, both stochastic and non-stochastic [Zhang and Sra, 2016], and even purely first-order accelerated methods [Zhang and Sra, 2018]. When it comes to retractions, rates of convergence have been proved in the Lipschitz setting for first and second-order methods [Boumal et al., 2016].

**Trivialization.**   The trick of parametrizing a Lie group with elements in the Lie algebra through the Lie exponential map has been commonly used under the name of *trivialization* in the area of differential equations on manifolds [Magnus, 1954, Iserles and Nørsett, 1999, Iserles et al., 2000]. We borrow the term, as the general idea behind these methods and ours is rather similar.

**Optimization through parametrizations.**   Parametrizing a manifold in terms of a Euclidean space is a common technique in optimization and machine learning. For example when doing computations

on symmetric positive definite matrices [Arsigny et al., 2006, 2007], compact Lie groups [Lezcano-Casado and Martínez-Rubio, 2019], the special orthogonal group [Helfrich et al., 2018] or the unitary group [Jing et al., 2017, Maduranga et al., 2018]. In [Dreisigmeyer, 2018], it is used through the Riemannian exponential to adapt $0^{\text{th}}$ order methods to naturally reductive homogeneous manifolds.

Our work finds the closest connections in the papers [Lezcano-Casado and Martínez-Rubio, 2019, Helfrich et al., 2018, Maduranga et al., 2018] These papers present the use of the Lie exponential and the Cayley map for optimization on $\mathrm{SO}(n)$. Our framework can be seen as an extension that can be implemented on top of them at a negligible execution cost. We also show that this theoretical improvement translates into a better convergence in practice in Section 7.

# 3 Problem Set-Up

We include a short introduction to the concepts used from differential and Riemannian geometry in Appendix A.

We are interested in approximating the following problem over a connected manifold $\mathcal{M}$

$$\min_{x \in \mathcal{M}} f(x).$$

A differentiable manifold does not carry intrinsically any metric information. As such, if one is interested in talking about concepts like the distance to the optimum, or the steepest descent direction, it is necessary to put additional structure on the problem. One way to do this is to consider a Riemannian metric $g$ on $\mathcal{M}$, turning $\mathcal{M}$ into a Riemannian manifold.

## 3.1 The classic approach: Riemannian gradient descent

Given a complete metric on $\mathcal{M}$, we can define geodesics $\gamma_{p,v} \colon [0, \infty) \to \mathcal{M}$ such that $\gamma_{p,v}(0) = p$, $\gamma'_{p,v}(0) = v$ for $v \in T_p\mathcal{M}$. Then, the Riemannian exponential map is defined simply as the map that maps rays starting at the origin in the tangent space to geodesics on $\mathcal{M}$. In symbols, $\exp_p(tv) \coloneqq \gamma_{p,v}(t)$ for $t \geq 0$.

Using the Riemannian exponential, one can define Riemannian gradient descent in an analogous way to the Euclidean case:

$$x_{t+1} = \exp_{x_t}(-\eta \nabla f(x_t)).$$

In plain words, the algorithm follows the geodesic in the direction of steepest descent $-\nabla f(x_t)$ for a time $\eta > 0$. This approach has been extensively studied in the literature and it has been proven to enjoy similar convergence properties to its Euclidean counterpart [Absil et al., 2009, Bonnabel, 2013, Boumal et al., 2016, Zhang et al., 2016].

Sometimes it is convenient, due to computational constraints, to use a first order approximation to the exponential rather than the exponential map. This idea is encapsulated in the concept of a retraction.

**Definition 3.1 (Retraction).** A differentiable map $r \colon T\mathcal{M} \to \mathcal{M}$ is called a retraction if for every $p \in \mathcal{M}$, the map $r_p \colon T_p\mathcal{M} \to \mathcal{M}$ satisfies $r_p(0) = p$ and $(\mathrm{d}r_p)_0 = \mathrm{Id}$.

The update rule of Riemannian gradient descent along a retraction $r$ is then given by

$$x_{t+1} = r_{x_t}(-\eta \nabla f(x_t)).$$

In many cases, this update rule is enough to recover the same convergence guarantees as in Riemannian gradient descent along the exponential map [Boumal et al., 2016].

The main problem of Riemannian gradient descent comes from a practical point of view. On many practical problems, it has been empirically proved that algorithms like ADAM [Kingma and Ba, 2014], ADAGRAD [Duchi et al., 2011] or RMSPROP [Tieleman and Hinton, 2012] outperform vanilla SGD. These algorithms were designed to work on $\mathbb{R}^n$, and although generalizations for product manifolds are in order [cf., Becigneul and Ganea, 2019], it is not clear how to generalize them to most manifolds used in practice, and thus take advantage of them in the Riemannian setting.

# 4 Trivializations

We now introduce trivializations. Trivializations are functions that allow us to transform a constrained problem on a manifold to an unconstrained one.

**Definition 4.1 (Trivialization).** Given a manifold $\mathcal{M}$, we define a trivialization as a surjective map

$$\phi\colon \mathbb{R}^n \to \mathcal{M}.$$

**Example 4.2.** The most simple examples are found when $\mathcal{M}$ has a product structure, *i.e.*, for vectors. For example, for a fixed $n > 0$, consider component-wise functions like rectified linear units, parametrizing non-negative vectors $\mathrm{relu}\colon \mathbb{R}^n \to (\mathbb{R}^+)^n$ or the sigmoid function $\sigma\colon \mathbb{R}^n \to (0,1)^n$.

Having a trivialization in hand, we can transform a constrained optimization problem into an unconstrained one by composing $f$ with $\phi$.

$$\min_{y\in\mathbb{R}^n} f(\phi(y)).$$

**Remark.** When considering a parametrization $\phi$, the gradient $\nabla f(x)$ changes into the gradient $\nabla (f \circ \phi)(y)$ for $x = \phi(y)$. For a 1-dimensional trivialization, by the chain rule, if $\phi'(y) = 0$ for many $y \in \mathbb{R}$, $\phi$ will not be a good parametrization, because then $\nabla(f \circ \phi)(y) = \nabla f(\phi(y))\phi'(y) = 0$, even though $\nabla f(x)$ might not be zero. As such, not all trivializations are equally good.

We formalize this intuition for general trivializations in the following theorem.

**Theorem 4.3.** *Let $\phi\colon \mathbb{R}^n \to \mathcal{M}$ be a diffeomorphism. Then, solving the problem $\min_{y\in\mathbb{R}^n} f(\phi(y))$ through gradient descent accounts for solving the problem $\min_{x\in\mathcal{M}} f(x)$ using Riemannian gradient descent for a certain metric on $\mathcal{M}$ induced by $\phi$.*

*Proof.* See Appendix B. $\qquad\qquad\qquad\qquad\qquad\qquad\qquad\qquad\qquad\qquad\qquad\qquad\qquad\Box$

This result tells us that, if $\phi$ is a diffeomorphism, $\phi$ will not add local minima or saddle points. It will simply act as a change of metric on the manifold. This already explains the good behavior of the $\tanh$ and sigmoid functions present in an LSTM or GRU in the context of gating.

At first sight, the situation of $\phi$ being a diffeomorphism seems too restrictive for general manifolds. We now introduce two parametrizations that are diffeomorphisms in *almost all* the manifold.[2]

## 4.1 The Riemannian trivialization

Consider now the Riemannian exponential map. By the Hopf-Rinow theorem, it is surjective whenever $(\mathcal{M}, g)$ is connected and complete. As such, in these cases, for any point $p \in \mathcal{M}$, the Riemannian exponential map $\exp_{\mathcal{M},p}\colon T_p\mathcal{M}(\cong \mathbb{R}^n) \to \mathcal{M}$ is an example of a trivialization.

**Geometric intuition about the Riemannian trivialization.** A direct corollary of Gauss' lemma says that the metric induced by the exponential parametrization $\exp_{\mathcal{M},p}$ is a first order approximation to the metric on the manifold around the point $p$ [Petersen, 2016, Lemma 5.5.7]. In other words, the Riemannian trivialization changes the metric into a new one with the square of the distance to $p$ for points near $p$.

Let us now look at the behavior of the Riemannian trivialization in global terms.

**Theorem 4.4 (Properties of the Riemannian trivialization).** *Let $(\mathcal{M}, g)$ be a connected, complete Riemannian manifold. Fix a point $p \in \mathcal{M}$. Let $U_p \subseteq T_p\mathcal{M}$ be the largest radially convex open neighborhood of zero on which $\exp_{\mathcal{M},p}$ is a diffeomorphism[3] then, $\exp_{\mathcal{M},p}(\overline{U}_p) = \mathcal{M}$.*

*Furthermore, define the* cut locus *in $T_p\mathcal{M}$ as $\tilde{C}_p := \overline{U}_p \backslash U_p$. If $V \in T_p\mathcal{M}$ is another open neighborhood of the origin that contains a point in $\tilde{C}_p$, then $\exp_{\mathcal{M},p}$ is not a diffeomorphism on $V$.*

*Proof.* See Section 5.7.3 in [Petersen, 2016]. $\qquad\qquad\qquad\qquad\qquad\qquad\qquad\qquad\qquad\Box$

Theorem 4.4 combined with Theorem 4.3 tell us that there exists a radially convex neighborhood of zero on which $\exp_{\mathcal{M},p}$ acts as a change of metric, and that $\exp_{\mathcal{M},p}$ stops being a diffeomorphism in the boundary—and hence, can add minima or saddle points at these points. As the image of $\overline{U}_p$ is the whole $\mathcal{M}$, if we write $C_p := \exp_{\mathcal{M},p}(\tilde{C}_p)$, we have that $\mathcal{M}$ decomposes in the disjoint union of $\exp_{\mathcal{M},p}(U_p)$ and $C_p$. The set $C_p$ is called the *cut locus at p*.

The cut locus is a remarkably slippery object of study given that, in general, it is not differentiable. Nonetheless, we can still measure the relative size of this set in a topological sense, by means of the Hausdorff dimension.

**Theorem 4.5 (Itoh and Tanaka [1998]).** *Let $\mathcal{M}$ be a connected and complete Riemannian manifold of dimension $n$. For a point $p \in \mathcal{M}$ the Hausdorff dimension of $\tilde{C}_p$ is either $0$ or $n-1$, and the Hausdorff dimension of $C_p$ is an integer less than $n$.*

Putting this result in the more familiar language of measures, we can argue that, although the cut locus can introduce problems in practice, the problematic set is not too large.[4]

**Corollary 4.6.** *$\tilde{C}_p$ has Lebesgue measure zero on $T_p\mathcal{M}$.*

*Proof.* By the definition of Hausdorff dimension, a set of dimension $n-1$ has $n$-Hausdorff measure $0$. Finally, just note that the $n$-Hausdorff measure is a multiple of the Lebesgue measure. $\square$

## 4.2 The Lie trivialization

We now introduce a useful trivialization for Lie groups and other matrix manifolds. Recall that for a Lie group $G$ we define its *Lie algebra* as the tangent space to the identity element $\mathfrak{g} := T_e G$. In Lie group theory there is a canonical trivialization given by the *Lie exponential*. For matrix Lie groups, which are the groups that we are interested in, the Lie exponential is exactly the exponential of matrices. We will denote the exponential of a matrix $A$ as $\exp(A)$ or $e^A$ for short.

For connected and compact Lie groups—*e.g.*, $\mathrm{SO}(n), \mathrm{U}(n), \mathrm{SU}(n), \mathrm{Sp}(n)$—this map is surjective and it coincides with the Riemannian trivialization at the identity for a suitable metric. If it is not surjective, we can still use it as a trivialization of the image of $\mathfrak{g}$ under $\exp$. In Section 6.2 we explain how to use the exponential parametrization in the whole Lie group, even when it is not surjective. Trivializations of this form for compact Lie groups were already studied in Lezcano-Casado and Martínez-Rubio [2019].

The following theorem is a generalization for matrix Lie groups of a classic result.

**Theorem 4.7 (Properties of the Lie exponential).** *Let $G$ be a matrix Lie group, the Lie exponential is a diffeomorphism on the set $U = \{A \in \mathfrak{g} \mid |\mathrm{Im}(\lambda_i(A))| < \pi\}$ with $\lambda_i(A)$ the eigenvalues.*

*Proof.* See Appendix C. $\square$

This result is the counterpart of Theorem 4.4 for the Lie trivialization on general matrix Lie groups. The boundary of this set has similar properties as those of the cut locus for the Riemannian trivialization for groups like $\mathrm{GL}(n)$ or $\mathrm{SO}(n)$.[5] As such, this trivialization presents the same problem as the Riemannian trivialization: It works as a change of metric for points that are close to the identity matrix, but it creates local minima and saddle points on some points of the manifold, which we might encounter as the optimization method progresses.

## 5 Dynamic Trivializations

In the last section, we have seen rather general families of trivializations that cover most of the manifolds used in practice. We have seen how these trivializations act as a change of metric around the initial point—$p$ in the case of the Riemannian trivialization and the identity matrix in the case of the Lie trivialization—but we have also shown that the optimization process can be affected as it deviates from the initial point.

Note that, in the case of the exponential trivialization, we have a map from any tangent space of $\mathcal{M}$ onto $\mathcal{M}$, but we are just using one of them as a trivialization. We can leverage the structure of $T\mathcal{M}$ in order to solve the problem that the trivializations introduced above. Instead of always using $\exp_{\mathcal{M},p}$, we can use it just for $K$ optimization steps and then change $p$ to the point on which we find ourselves on the manifold after those $K$ steps. This idea is formalized in the following algorithm.

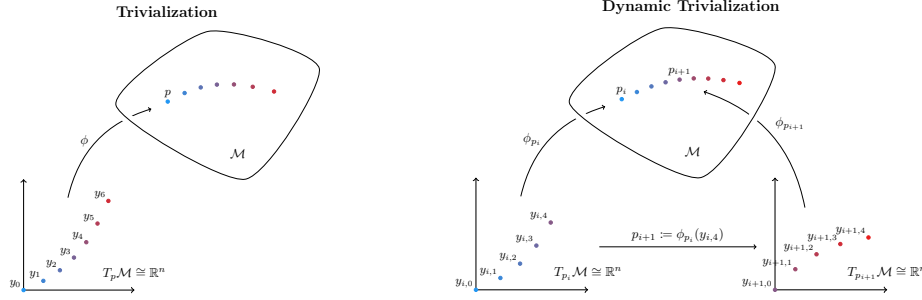

Figure 1: Example of the trivialization and dynamic trivialization procedure for $K = 4$.

**Algorithm 5.1 (Dynamic trivialization through retractions).** *Given a retraction $r$, an integer $K > 0$ or $K = \infty$, and a starting point $p_0$, the dynamic trivialization induced by $r$ is defined as the sequence of problems indexed by $i = 0, 1, \ldots$*

$$\min_{y \in T_{p_i} \mathcal{M}} f(r_{p_i}(y))$$

*where $p_{i+1} := r_{p_i}(y_{i,K}) \in \mathcal{M}$, and $y_{i,k} \in T_{p_i} \mathcal{M}$ for $k = 1, \ldots, K$, is a sequence of approximations given by a Euclidean optimization algorithm—e.g.,* SGD, ADAM, ADAGRAD, RMSPROP, ...—*applied to the $i$-th problem with starting point $y_{i,0} = 0$. We say that $p_i$ is the* basis *at step $i$.*

**Remark.** Note that in this case we have dropped the condition of $r_p \colon T_p \mathcal{M} \to \mathcal{M}$ being surjective. This is because, as long as $\mathcal{M}$ is connected, we can still reach any point in $\mathcal{M}$ in the optimization process by changing the basis of the dynamic trivialization whenever $K < \infty$.

This procedure has two interesting limit cases.

**Generalization of trivializations.** For $K = \infty$, *i.e.*, no change of basis, it reduces to the trivialization algorithms described in Section 4 with the trivialization $r_{p_0}$, provided that $r_{p_0}$ is surjective.

**Generalization of Riemannian gradient descent.** In the case $K = 1$, we are changing the basis of the trivialization on every step. When the optimization process used to generate the iterates $y_{i,k}$ is regular SGD, this method recovers exactly stochastic Riemannian gradient descent using $r$ as a retraction. For this, just note that by the chain rule and the definition of a retraction

$$\mathrm{d}(f \circ r_{p_i})_0 = (\mathrm{d}f)_{r_{p_i}(0)} \circ (\mathrm{d}r_{p_i})_0 = (\mathrm{d}f)_{r_{p_i}(0)} = (\mathrm{d}f)_{p_i}.$$

From this it follows that

$$\nabla(f \circ r_{p_i})(0) = \nabla f(p_i)$$

so the update rule simplifies for a learning rate $\eta > 0$ can be rewritten as

$$y_{i,1} = -\eta \nabla f(p_i) \qquad p_{i+1} = r_{p_i}(-\eta \nabla f(p_i))$$

and $p_{i+1}$ are exactly the iterates given by doing Riemannian stochastic gradient descent along the retraction $r$.

In particular, we have proved that for $r = \exp_{\mathcal{M}}$, we recover stochastic Riemannian gradient descent. As such, we can see dynamic trivializations as an interpolation between the trivialization method using $\exp_{\mathcal{M}}$ and stochastic Riemannian gradient descent.

More interesting is perhaps the case when we use a different optimizer to generate the iterates $y_{i,k}$. In this case, dynamic trivializations yield a natural generalization to manifolds of the algorithm used to generate the iterates, *i.e.*, ADAM, ADAGRAD, RMSPROP, *etc*.

## 6 Gradient Computations and Examples

The last missing piece needed to implement dynamic trivializations is the explicit computation of their gradients. We will do so for the two families presented above.

## 6.1 The matrix exponential

We first look at the matrix exponential. This function not only defines the Lie trivialization, but it is also essential to compute the Riemannian exponential in many matrix manifolds (*cf.*, Appendix E). In order to implement the dynamic trivialization algorithm within the context of first-order methods we need an approximation of the trivialization map and its gradient.

The current fastest machine-precision approximation to the matrix exponential was formulated in Al-Mohy and Higham [2009b]. On the other hand, it is not clear how to compute the gradient of this parametrization. The following proposition settles this problem.

**Proposition 6.1 (Gradient of the exponential parametrization).** *Let $f \colon \mathbb{R}^{n \times n} \to \mathbb{R}$ be a function defined on matrices, and let* $\exp$ *be the matrix exponential, we have that*

$$\nabla(f \circ \exp)(A) = (\mathrm{d}\exp)_{A^\intercal}(\nabla f(e^A)).$$

*Proof.* See Appendix D. □

This proposition together with the approximation algorithm for $\mathrm{d}\exp$ presented in Al-Mohy and Higham [2009a] allows us to approximate to machine-precision this gradient.

This formula readily allows for the implementation of the Riemannian dynamic trivialization on many matrix manifolds. We give examples of some of these in Appendix E.

## 6.2 Lie exponential for matrix Lie groups

The Lie exponential on a Lie group $G$ is just defined on the Lie algebra $\mathfrak{g} = T_e G$. On matrix Lie groups, we can identify any tangent space of $G$ with $\mathfrak{g}$. Explicitly, if $\tilde{A} \in T_B G$, then $B^{-1}\tilde{A} \in \mathfrak{g}$. Furthermore, if we choose a left-invariant metric on the Lie group, we can then use left multiplication to map the result $\exp(B^{-1}\tilde{A})$ to a neighborhood of $B$. In symbols, we can define

$$\exp_B \colon T_B G \to G$$
$$\tilde{A} \mapsto B \exp(B^{-1}\tilde{A})$$

We give the gradient of this parametrization in Corollary D.3. This function constitutes a dynamic trivialization on any connected matrix Lie group, like, for example, $\mathrm{SO}(n)$, $\mathrm{U}(n)$, $\mathrm{SL}(n)$, or $\mathrm{GL}^+(n)$.

## 6.3 Other retractions

Sometimes one cannot afford to approximate the exponential exactly, as it can be very costly. In this case, the standard alternative are retractions Boumal et al. [2016].

**Cayley map.** This is one of the most well known retractions to optimize over $\mathrm{SO}(n)$ (*cf.*, Absil et al. [2009], Helfrich et al. [2018])

$$\mathrm{cay} \colon \mathrm{Skew}(n) \to \mathrm{SO}(n)$$
$$A \mapsto (\mathrm{I} + A)(\mathrm{I} - A)^{-1}$$

This can be made into a dynamic retraction using the same trick as we did with the exponential, considering $\mathrm{cay}_B(A) = B \, \mathrm{cay}(B^{-1}\tilde{A})$, for $B \in \mathrm{SO}(n)$, $\tilde{A} \in T_B \, \mathrm{SO}(n)$.

**Projectors.** Another common retraction used in matrix manifolds $\mathcal{M} \subseteq \mathbb{R}^{n \times n}$ is the one given by $\pi_{\mathcal{M}}(x + v)$ for $x \in \mathcal{M}$, $v \in T_x \mathcal{M}$ and $\pi_{\mathcal{M}}$ the projection from $\mathbb{R}^{n \times n}$ onto $\mathcal{M}$. For example, for $\mathcal{M} = \mathrm{SO}(n)$, we have that for a matrix $B \in \mathbb{R}^{n \times n}$ with SVD decomposition $B = U\Sigma V^\intercal$, its projection onto $\mathrm{SO}(n)$ is given by $\pi_{\mathrm{SO}(n)}(B) = UV^\intercal$.[6] with gradient computed in Kenney and Laub [1991, Eq. 2.18].

We workout more useful examples for common manifolds in Appendix E.

Table 1: Best test accuracy at MNIST and P-MNIST.

| MODEL | N | MNIST | P-MNIST |
|---|---|---|---|
| DTRIV1 | 170 | **98.3** | **95.2** |
| DTRIV100 | 170 | 98.2 | 95.1 |
| DTRIV∞ | 170 | 98.1 | 95.0 |
| EXPRNN | 170 | 98.0 | 94.9 |
| SCORNN | 170 | 97.2 | 94.8 |
| SCURNN | 116 | 97.6 | 94.9 |
| LSTM | 128 | 81.9 | 79.5 |
| RGD | 116 | 94.7 | 92.5 |
| DTRIV1 | 360 | 98.4 | 96.3 |
| DTRIV100 | 360 | 98.8 | 96.4 |
| DTRIV∞ | 360 | **98.9** | **96.5** |
| EXPRNN | 360 | 98.4 | 96.2 |
| SCORNN | 360 | 98.1 | 95.9 |
| SCURNN | 250 | 98.3 | 96.2 |
| LSTM | 256 | 88.8 | 88.8 |
| RGD | 256 | 96.1 | 93.9 |
| DTRIV1 | 512 | 98.7 | 96.7 |
| DTRIV100 | 512 | **99.1** | 96.7 |
| DTRIV∞ | 512 | 99.0 | **96.8** |
| EXPRNN | 512 | 98.7 | 96.6 |
| SCORNN | 512 | 98.2 | 96.5 |
| LSTM | 512 | 91.9 | 91.8 |
| RGD | 512 | 97.3 | 94.7 |

Table 2: Test MSE at the end of the epoch with the lowest validation MSE for the TIMIT task.

| MODEL | N | VAL. MSE | TEST MSE |
|---|---|---|---|
| DTRIV1 | 224 | 6.55 | 6.54 |
| DTRIV100 | 224 | 4.80 | 4.77 |
| DTRIV∞ | 224 | **4.75** | **4.71** |
| EXPRNN | 224 | 5.34 | 5.30 |
| SCORNN | 224 | 9.26 | 8.50 |
| SCURNN | 128 | 9.42 | 7.23 |
| LSTM | 84 | 15.42 | 14.30 |
| RGD | 128 | 15.07 | 14.58 |
| DTRIV1 | 322 | 4.56 | 4.55 |
| DTRIV100 | 322 | 3.80 | 3.76 |
| DTRIV∞ | 322 | **3.39** | **3.76** |
| EXPRNN | 322 | 4.42 | 4.38 |
| SCORNN | 322 | 8.48 | 7.82 |
| LSTM | 120 | 13.93 | 12.95 |
| RGD | 192 | 15.10 | 14.50 |
| DTRIV1 | 425 | 4.21 | 4.17 |
| DTRIV100 | 425 | 2.02 | 1.99 |
| DTRIV∞ | 425 | **2.00** | **1.97** |
| EXPRNN | 425 | 5.52 | 5.48 |
| SCORNN | 425 | 7.97 | 7.36 |
| SCURNN | 258 | 4.40 | 3.39 |
| LSTM | 158 | 13.66 | 12.62 |
| RGD | 256 | 14.96 | 14.69 |

# 7 Experiments

In this section, we assess the effectiveness of dynamic trivializations (DTRIV) in the context of orthogonal optimization. We test the framework with the basis changed every $K = 1, 100, \infty$ steps.

We compare it against the most performant previous approaches presented for this task in the context of orthogonal optimization and a vanilla LSTM. These approaches are orthogonal exponential trivialization [EXPRNN Lezcano-Casado and Martínez-Rubio, 2019], orthogonal and unitary Cayley trivializations [SCORNN / SCURNN Helfrich et al., 2018, Maduranga et al., 2018], and Riemannian gradient descent [RGD Wisdom et al., 2016].

The architecture on which we are testing the dynamic trivialization is the same as in the papers above: A vanilla RNN with an orthogonal layer parametrized using the Lie trivialization (*cf.*, Section 6.2)

$$h_{t+1} = \sigma(\exp_B(A)h_t + Tx_{t+1}).$$

The update procedure for $B$ was described in Algorithm 5.1 ($K = 1, 100, \infty$).

**Remark.** Note that RGD is equivalent to DTRIV1 together with the optimizer SGD. Furthermore, EXPRNN is equivalent DTRIV∞ only that EXPRNN has the basis on the identity matrix and DTRIV∞ has the basis on the matrix to which it is initialized.

We test this architecture on two different tasks that have become the standard to test the performance of RNNs in the context of long-term recall and long-term memory, namely the pixel-by-pixel MNIST and the TIMIT dataset [Arjovsky et al., 2016, Henaff et al., 2016, Mhammedi et al., 2017, Helfrich et al., 2018, Maduranga et al., 2018, Lezcano-Casado and Martínez-Rubio, 2019]. We do not present results for the copying problem, as task is too simple to draw any meaningful conclusions, as explained in Henaff et al. [2016]. [7]

We detail all the hyperparameters and set-up in Appendix F. The code and instructions to replicate these experiments can be found in

https://github.com/Lezcano/expRNN

### 7.1 Pixel-by-pixel MNIST

This task consists of classifying the hand-written images of numbers in the MNIST dataset [LeCun and Cortes, 2010] by processing them as a sequence pixel-by-pixel. Each image has $28 \times 28$ pixels, so the sequences are of length $784$. The *unpermuted task* (MNIST) processes the row-by-row flattened image, the *permuted task* (P-MNIST) samples a permutation of size $784$ at the beginning and then uses it to permute all the images after flattening them. This task was introduced in Le et al. [2015].

Table 1 is structured so that architectures with the same number of parameters are compared together. As we can see, the addition of any dynamic trivialization to the Lie parametrization improves the results on this experiment by $0.4\%$ out of the $1.3\%$ possible in the largest size. Moreover, it always improves the previous results, suggesting that it is always a better option to use dynamic trivializations rather than just plain trivializations. In general, we saw that DTRIV100 and DTRIV$\infty$ gave the highest stability and the best results across the experiments.

### 7.2 TIMIT speech dataset

The TIMIT dataset [S Garofolo et al., 1992] is a set of variable-length real-world speech recordings. These recordings are first downsampled to 8kHz and then transformed into log-magnitudes via a short-time Fourier transform, giving sequences of $129$ complex numbers per step, and a variable length between $61$ and $490$. The task consists of predicting the next log-magnitude given the previous ones. This experiment was introduced in Wisdom et al. [2016].

In this experiment we see a similar behavior of the dynamic trivializations as the one already seen in the MNIST and P-MNIST experiments. It also happens in this experiment that DTRIV100 and DTRIV$\infty$ always improve the performance of their static counterparts with base at the identity and of RGD.

In the experiments in SCURNN they explicitly mention that they are computing the MSE without discarding the zeros used to pad the variable-length sequences [Maduranga et al., 2018]. As such, when computing the MSE, they are dividing by an incorrect number—the longest element in the batch times the elements in the batch—rather than by the correct one—the sum of the lengths of all the elements in the batch. We computed the correct validation and test loss in Table 2.

## 8 Conclusion and Future Work

In this paper we have presented a novel way to perform optimization on manifolds that combines the strengths of the two most popular optimization techniques used in machine learning and neural networks—parametrizations and Riemannian gradient descent. We have shown that, by moving the initial point of the parametrization, as the metric is distorted less from the Euclidean one, we can achieve an improvement on the convergence of the neural network.

We leave open an interesting line of research based on applying dynamic trivializations to allow optimization on other interesting manifolds. As a first step in this direction, we detail examples of some computations for the most common manifolds used in optimization in Appendix E.

## Acknowledgements

We would like to thank the help of Jaime Mendizabal and Momchil Konstantinov for the very useful feedback and suggestions and Prof. Andras Juhasz for the computing power.

The work of MLC was supported by the Oxford-James Martin Graduate Scholarship and the "la Caixa" Banking Foundation (LCF/BQ/EU17/11590067).

## Footnotes

[1] An implementation can be found at: `https://github.com/Lezcano/expRNN`

[2]This is taken with respect to the canonical Borel measure on the manifold induced by the metric.

[3]A more formal way to define it would be $\overline{U}_p := \{v \in T_p\mathcal{M} \mid \exp_p(tv)$ is length minimizing for $t \in [0,1]\}$.

[4]The analogous result for $C_p$ with respect to the Borel measure induced by the volume form is also true.

[5]The constant $\pi$ is tight for matrix manifolds that contain matrices with eigenvalues that are $2\pi i$ apart. For these manifolds, the matrix exponential fails to be a diffeomorphism on some points of the boundary of $U$.

[6]Formally, $\pi_{\mathrm{SO}(n)}$ is well-defined for matrices such that $\det B > 0$, that is, $\pi_{\mathrm{SO}(n)} \colon \mathrm{GL}^+(n) \to \mathrm{SO}(n)$. Note that this function is not a diffeomorphism but a submersion. Theorem 4.3 can be extended to this case.

[7]For reference, dynamic trivializations are also able to converge to the correct answer stably, as EXPRNN.

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
