[Supplementary Material]

# A  Differential and Riemannian Geometry

In this section we give a short introduction to the concepts used in the paper and in the appendix of the theories of differential and Riemannian geometry and Lie groups. The standard modern introduction to differential geometry is Lee [2013]. This book also gives an introduction to Lie groups. Introductory texts in Riemannian geometry are do Carmo [1992], Lee [2018]. Introductory references for Lie groups are Rossmann [2006], Hall [2015]. Although not covered in this summary, two more advanced texts that cover the classical theory of the cut locus through Jacobi fields are Gallot et al. [2012], Petersen [2016].

## A.1  Differential Geometry

Let $\mathcal{M}$ be an $n$-dimensional differentiable real manifold. $\mathcal{M}$ has an associated global object called the *tangent bundle* $T\mathcal{M} := \sqcup_{p \in \mathcal{M}} \{p\} \times T_p\mathcal{M}$, that is, the disjoint union of all the tangent spaces at every point of $\mathcal{M}$. The tangent bundle comes with a structure of a $2n$-dimensional differentiable manifold. A point in $T\mathcal{M}$ consists then of a pair $(p, v)$ with $p \in \mathcal{M}$ and $v \in T_p\mathcal{M}$. On each point, we also have the *cotangent space* $T_p^*\mathcal{M}$ of linear applications from vectors onto the real numbers. The disjoint union of all the cotangent spaces is another manifold $T^*\mathcal{M}$ called the *cotangent bundle*. When considering these bundles, tangent spaces $T_p\mathcal{M}$ and cotangent spaces $T_p^*\mathcal{M}$ are sometimes called *fibres*.

An *affine connection* $\nabla$ is a bilinear form that, given two vector fields $X, Y$, assigns a new one $\nabla_X Y$, and it is tensorial on the first component and Leibnitz on the second. An affine connection defines a notion of *parallel vector fields*. We say that a vector field $Z$ is parallel along a curve $\gamma \colon [0, 1] \to \mathcal{M}$ if $\nabla_{\gamma'} Z = 0$ where $\gamma' := \mathrm{d}\gamma(\frac{\mathrm{d}}{\mathrm{d}t})$. For any curve, given an initial vector $Z_0$, there exists a unique parallel vector field $Z$ along it such that $Z(0) = Z_0$. We say that the vector $Z(t)$ is the parallel transport of $Z(0)$ for $t \in [0, \varepsilon)$.

## A.2  Riemannian Geometry

A *Riemannian manifold* is a differentiable manifold together with a smooth metric $g_p \colon T_p\mathcal{M} \times T_p\mathcal{M} \to \mathbb{R}$ which is symmetric and positive definite. A metric induces a distinguished connection called the *Levi-Civita connection*. This is the unique connection that is torsion-free, $\nabla_X Y - \nabla_Y X = [X, Y] := XY - YX$, and it is compatible with the metric, $\mathrm{D}_Z(g(X, Y)) = g(\nabla_Z X, Y) + g(X, \nabla_Z Y)$, where $\mathrm{D}_Z$ denotes the directional derivative in the direction of $Z$. Whenever we talk about a connection on a Riemannian manifold we will always be referring to the Levi-Civita connection.

A Riemannian manifold has a notion of length of a differentiable curve $c \colon [0, 1] \to \mathcal{M}$, $L(c) = \int_0^1 \|\gamma'(t)\| \, \mathrm{d}t$. When the manifold is connected, this allows to put the structure of a metric space on the manifold, defining the distance between two points as the length of the shortest piece-wise differentiable curve joining these two points.

Given a connection, we define a *geodesic* $\gamma \colon [0, \varepsilon) \to \mathcal{M}$ as a self-parallel curve, $\nabla_{\gamma'} \gamma' = 0$. Geodesics are defined for any starting conditions $(p, v) \in T\mathcal{M}$, $\gamma(0) = p$, $\gamma'(0) = v$ on an interval $[0, \varepsilon)$. If a Riemannian manifold is connected and complete, the *Hopf-Rinow theorem* asserts that geodesics not only exist locally, but globally, that is, they can be extended indefinitely taking $\varepsilon = \infty$ giving $\gamma \colon [0, \infty) \to \mathcal{M}$. Furthermore, Hopf-Rinow adds that, under the same conditions, there exists a geodesic connecting any two given points. When the connection comes from a metric, geodesics are the locally length-minimizing curves on $\mathcal{M}$.

Given a connection, we define the exponential map as $\exp_p(v) := \gamma_{p,v}(1)$ where $\gamma_{p,v}$ is the geodesic with initial conditions $(p, v)$. On a connected and complete Riemannian manifold, Hopf-Rinow says that the exponential map is defined in the whole tangent bundle.

A metric induces an isomorphism between the tangent and cotangent bundle $\alpha \colon T\mathcal{M} \to T^*\mathcal{M}$ defined as $\alpha(X) := g(X, -)$. $\alpha$ is sometimes called the *musical isomorphism*. The gradient of a function is defined as the vector field associated to the differential form $\mathrm{d}f$ through this isomorphism $\nabla f := \alpha^{-1}(\mathrm{d}f)$. In other words, it is the vector field such that $\mathrm{d}f = g(\nabla f, -)$. As such, the gradient depends on the choice of metric. A metric also allows to define the adjoint of a differential

$d\phi\colon T_p\mathcal{M} \to T_{\phi(p)}\mathcal{M}$ at a point $p \in \mathcal{M}$ as the application $d\phi^*\colon T_{\phi(p)}\mathcal{M} \to T_p\mathcal{M}$ such that for every $X \in T_p\mathcal{M}, Y \in T_{\phi(p)}\mathcal{M}$ we have that $g(d\phi(X), Y)_{\phi(p)} = g(X, d\phi^*(Y))_p$.

## A.3 Lie groups

A Lie group $G$ is a differentiable manifold equipped with a differentiable group structure. Lie groups have a distinguished tangent space called the *Lie algebra*, which is the tangent space at the identity $\mathfrak{g} := T_e G$. Any closed subgroup of a Lie group is itself a Lie group. A (real) *matrix manifold* is a closed subgroup of the *general linear group* $\mathrm{GL}(n) = \{B \in \mathbb{R}^{n \times n} \mid \det A \neq 0\}$. The Lie algebra of the general linear group is $\mathfrak{gl}(n) = \mathbb{R}^{n \times n}$. In general, the general linear group of a vector space $V$ is the Lie group formed by the invertible automorphisms of $V$, $\mathrm{GL}(V)$.

On a Lie group, one has for every $g, x \in G$ the diffeomorphisms given by left translations $L_g(x) := gx$, right translations $R_g(x) := xg$, and conjugation $c_g(x) = gxg^{-1}$. Using left translations, one can identify any tangent space with the Lie algebra via the vector space isomorphism $(dL_{g^{-1}})_g\colon T_gG \to \mathfrak{g}$. The differential of the conjugation at the identity is called the *adjoint representation of $G$*, $\mathrm{Ad}\colon G \to \mathrm{GL}(\mathfrak{g})$. The differential of $\mathrm{Ad}$ at the identity is the *adjoint representation of $\mathfrak{g}$*, $\mathrm{ad}\colon \mathfrak{g} \to \mathrm{End}(\mathfrak{g})$. For matrix Lie groups, $\mathrm{Ad}_g(X) = gXg^{-1}$ and $\mathrm{ad}_X(Y) = [X, Y]$.

Given a vector $X \in \mathfrak{g}$, we can consider the *one parameter subgroup* with starting vector $X$, which is the unique group homomorphism $\gamma_X\colon \mathbb{R} \to G$ such that $\gamma_X'(0) = X$. The Lie exponential is then defined for every $X \in \mathfrak{g}$ as $\exp(X) := \gamma_X(1)$. For matrix Lie groups, the Lie exponential is given by the exponential of matrices.

A Riemannian metric on a Lie group is said to be left (resp. right) invariant if it turns left (resp. right) translations into isometries. A metric is said to be bi-invariant if it is both left and right invariant. Every Lie group admits a left-invariant metric, given by choosing any inner product in $\mathfrak{g}$ and pushing it forward using $L_{g^{-1}}$. Only compact Lie groups, commutative Lie groups, and products of them admit bi-invariant metrics. When a Lie group is equipped with a bi-invariant metric, the Lie exponential coincides with the Riemannian exponential at the identity.

# B Parametrizations on Manifolds

In this section we look at the problem of how does optimizing $f \circ \phi$ affect the optimization problem, depending on the properties of $\phi$. As a disclaimer we would like to mention that, although this section and next section are original, most of them would be considered routine in the field of differential geometry.

Consider the optimization problem

$$\min_{x \in \mathcal{M}} f(x) \tag{1}$$

where $\mathcal{M}$ is a Riemannian manifold. In this section we will look at parametrizations, which can be regarded as a generalization of certain trivializations, when the domain is not necessary $\mathbb{R}^n$ but a Riemannian manifold.

Suppose that we have access to a diffeomorphism between Riemannian manifolds

$$\phi\colon \mathcal{N} \to \mathcal{M}.$$

and denote the metric on $\mathcal{N}$ as $g_2$. We say that $\phi$ is a *parametrization of $\mathcal{M}$ in terms of $\mathcal{N}$*.

We can then consider the problem

$$\min_{y \in \mathcal{N}} f(\phi(y)).$$

In order to apply a first-order method to this new problem we first have to compute the gradient of this new function $f \circ \phi$. In order to do so, let us first define some notation.

Denote by $d\phi$ and $d\phi'$ the differential and its dual

$$d\phi\colon T\mathcal{N} \to T\mathcal{M}$$
$$d\phi'\colon T^*\mathcal{M} \to T^*\mathcal{N}$$

and denote by $\alpha$ and $\beta$ the canonical isomorphisms between the tangent and the cotangent bundle induced by the metrics

$$\alpha: T\mathcal{M} \xrightarrow{\cong} T^*\mathcal{M}$$

$$\beta: T\mathcal{N} \xrightarrow{\cong} T^*\mathcal{N}.$$

Finally, denote by $\mathrm{d}\phi^*$ the fibre-wise adjoint with respect to the two metrics of $\mathrm{d}\phi$

$$\mathrm{d}\phi^*: T\mathcal{M} \to T\mathcal{N}.$$

**Proposition B.1.** *Using the notation above, the following relation holds*

$$\beta \circ \mathrm{d}\phi^* = \mathrm{d}\phi' \circ \alpha.$$

*Proof.* For $Y \in T\mathcal{N}, X \in T\mathcal{M}$, we have that

$$(\mathrm{d}\phi' \circ \alpha)(X)(Y) = \alpha(\mathrm{d}\phi(Y))(X) = \beta(\mathrm{d}\phi^*(X))(Y) = (\beta \circ \mathrm{d}\phi^*)(X)(Y). \qquad \square$$

Using this proposition, we can compute the gradient with respect to the new parametrization.

**Corollary B.2.** *Let $\phi: \mathcal{N} \to \mathcal{M}$ be a smooth map between Riemannian manifolds and $f$ be a function on $\mathcal{M}$. We have that*

$$\nabla(f \circ \phi) = \mathrm{d}\phi^*(\nabla f).$$

*Proof.* This is direct using the previous proposition since

$$\nabla(f \circ \phi) := \beta^{-1}(\mathrm{d}(f \circ \phi)) = (\beta^{-1} \circ \mathrm{d}\phi')(\mathrm{d}f) = \mathrm{d}\phi^*(\nabla f). \qquad \square$$

This motivates the definition of the metric associated to a parametrization $\phi$.

**Definition B.3 (Metric associated to a parametrization).** A parametrization between Riemannian manifolds $\phi: \mathcal{N} \to \mathcal{M}$ induces a metric on $\mathcal{M}$ as per

$$(\phi_* g_2)(X_1, X_2)_p := g_2(\mathrm{d}\phi^*(X_1), \mathrm{d}\phi^*(X_2))_{\phi^{-1}(p)} \qquad \forall p \in \mathcal{M}.$$

This is a metric since $\mathrm{d}\phi^*(X) = 0$ if and only if $X = 0$ by the inverse function theorem, given that $\phi$ is a diffeomorphism.

Another way of looking at this construction is through the lens of submersions.

**Definition B.4 (Riemannian Submersion).** A Riemannian submersion is a surjective map $\phi: \mathcal{N} \to \mathcal{M}$ such that its differential is surjective at every point and

$$\mathrm{d}\phi: (\ker(\mathrm{d}\phi))^\perp \to T\mathcal{M}$$

is an isometry.

This is equivalent to saying that the adjoint $\mathrm{d}\phi^*$ should be an isometry. This is exactly the construction that we are using, we take the metric that converts $\phi$ into a Riemannian submersion.

We now look at this new metric. We will prove that doing gradient descent using a retraction along $\phi_* g_2$, is not a retraction with respect to $g_2$, and hence, it constitutes an optimization method fundamentally different to the original Riemannian gradient descent.

Using this metric, gradient descent on $\mathcal{M}$ with a step-size $\eta > 0$ is given by the map

$$y_{t+1} = (\phi \circ \exp_{\mathcal{N}, g_2} \circ \mathrm{d}\phi^*)(-\eta \nabla f(y_t))$$

where $\exp_{\mathcal{N}, g_2}: T\mathcal{N} \to \mathcal{N}$ is the Riemannian exponential map on $(\mathcal{N}, g_2)$. Note that since $\nabla f = \alpha^{-1} \circ \mathrm{d}f$, this step does not depend on the initial metric on $\mathcal{M}$, as we already observed in the proof of Corollary B.2.

More generally, recall the definition of a retraction.

**Definition B.5 (Retraction).** A differentiable map $r: T\mathcal{N} \to \mathcal{N}$ is called a retraction if for every $p \in \mathcal{N}$

$$r_p(0) = p \qquad \text{and} \qquad (\mathrm{d}r_p)_0 = \mathrm{Id}.$$

In other words, $r$ is an order one approximation to the Riemannian exponential.

As proved in Boumal et al. [2016], under Lipschitzness conditions, it is enough to follow retractions rather than the exponential map in order to achieve convergence to a local minimum with Riemannian gradient descent. As such, a natural question to ask is whether the function that defines the update step defines a retraction.

**Proposition B.6.** *Let $(\mathcal{M}, g_1), (\mathcal{N}, g_2)$ be Riemannian manifolds. Let $\phi$ be a parametrization between them and let $r \colon T\mathcal{N} \to \mathcal{N}$ be a retraction. The map*

$$\phi_* r := \phi \circ r \circ \mathrm{d}\phi^* \colon T\mathcal{M} \to \mathcal{M}$$

*is a retraction if and only if $\phi$ is a local isometry.*

*Proof.* It is clear that $(\phi_* r)_p(0) = p$. For the second condition, differentiating, we have that the map is a retraction if and only if

$$\mathrm{d}\phi \circ \mathrm{d}\phi^* = \mathrm{Id}_{T_p\mathcal{M}}.$$

or equivalently $\mathrm{d}\phi^{-1} = \mathrm{d}\phi^*$. Now,

$$(\mathrm{d}\phi \circ \mathrm{d}\phi^* \circ \mathrm{d}\phi \circ \mathrm{d}\phi^*)_p = \mathrm{Id}_{T_p\mathcal{M}}$$

so

$$(\mathrm{d}\phi^* \circ \mathrm{d}\phi)_{\phi^{-1}(p)} = (\mathrm{d}\phi^{-1} \circ \mathrm{d}\phi^*)_{\phi^{-1}(p)} = \mathrm{Id}_{T_{\phi^{-1}(p)}\mathcal{N}}.$$

Finally, since $\mathrm{d}\phi^*$ is the adjoint operator with respect to the metrics $g_2$ and $g_1$, evaluating this last expression on two points using the metric

$$g_1(\mathrm{d}\phi(u), \mathrm{d}\phi(v)) = g_2(u, v) \qquad \forall u, v \in T_{\phi^{-1}(p)}\mathcal{N},$$

which is equivalent to $\phi$ being a local isometry. $\square$

This is not a surprising result, since a retraction is a map that preserves the gradient. The way we have defined $\phi_* r$ is such that it preserves the gradient with respect to $g_2$. If it also preserved the gradient with respect to $g_1$, that would mean that the gradients with respect to the two metrics are the same, modulo a transformation through $\mathrm{d}\phi^*$, in other words, $\mathrm{d}\phi$ should be a local isometry.

## C  Proof of Theorem 4.7

In this section we generalize to general matrix Lie groups the classic proof presented in Theorem D.2. in Lezcano-Casado and Martínez-Rubio [2019].

In order to generalize this proof, we need the following theorem.

**Theorem C.1 (Theorem $4$ in Hille [1958]).** *Let $A, B \in \mathbb{C}^{n \times n}$. If there are no two eigenvalues in $A$ such that their difference is of the form $2n\pi i$ for $n > 0$ and, if $e^A = e^B$, we have that $A$ and $B$ commute.*

With this theorem in hand we can prove the following strengthened result.

**Theorem C.2 (Properties of the Lie exponential).** *Let $G$ be a closed subgroup of $\mathrm{GL}(n, \mathbb{C})$, the Lie exponential is a diffeomorphism on the set $U = \{A \in \mathfrak{g} \mid |\mathrm{Im}(\lambda_i(A))| < \pi\}$ with $\lambda_i(A)$ the eigenvalues of $A$.*

*Proof.* The fact that the differential of the exponential is surjective on this domain is classic (*cf.*, Section 1, Proposition 7 in Rossmann [2006]). As such, we just have to prove that the exponential is injective on this domain.

If $A \in U$ is diagonalizable, $A = C\Sigma C^{-1}$ with $\Sigma$ diagonal, and $\exp(A) = C \exp(\Sigma) C^{-1}$ where $\exp(\Sigma)$ is just the element-wise exponential of the diagonal elements.

By Hille [1958, Theorem 4], if two matrices $A, B \in \mathrm{GL}(n, \mathbb{C})$ such that $e^A = e^B$ do not have two eigenvalues that are $2n\pi i$ apart for $n \neq 0$, then $A$ and $B$ commute.

Any two matrices in $U$ have this property, so

$$e^A e^{-B} = e^{A-B} = \mathrm{I}.$$

As $A, B \in U$, $\mathrm{Im}(\lambda_i(A - B)) < 2\pi$, and as the eigenvalues of $e^{A-B}$ are 1, and the eigenvalues of the exponential of a matrix is the exponential of its eigenvalues, the eigenvalues of $A - B$ are all zero. Putting it in Jordan-normal form, we can assume that $A - B$ is upper triangular so, as the eigenvalues of $A - B$ are zero, we can assume that $A - B$ is also nilpotent.

Now, if we prove that the only upper triangular nilpotent matrix that is mapped to the identity matrix under the exponential is the null matrix, we finish the proof, as this would imply that $A = B$.

The set of upper-triangular nilpotent matrices is the Lie algebra of the Lie group of upper triangular matrices with ones on the diagonal. Recall the formula for the logarithm

$$\log(B) = \sum_{k=1}^{\infty} (-1)^{k+1} \frac{(B - \mathrm{I})^k}{k}.$$

Whenever $B$ is upper triangular with ones on the diagonal, $B - \mathrm{I}$ is nilpotent, so the series converges. As such, all these matrices have one and just one logarithm in $U$. In particular, the exponential is a bijection on this set. $\qquad\square$

## D  Gradient of the Matrix Exponential

In this section we give a formula for the gradient of the pullback of a function by the matrix exponential. The implementation of these formulas in practice and how can they be applied on different manifolds is considered in Appendix E.

We will prove a stronger result, which also applies to other matrix functions like $\cos(X)$, $\sin(X)$ and, with minor modifications, to functions like $\sqrt{X}$, $X^{1/n}$, and $\log(X)$.[8]

**Theorem D.1.** *Consider a real analytic function*

$$\phi \colon \mathbb{R} \to \mathbb{R}$$

$$x \mapsto \sum_{n=0}^{\infty} \frac{a_n}{n!} x^n$$

*with associated matrix function*

$$\phi \colon \mathbb{R}^{n \times n} \to \mathbb{R}^{n \times n}$$

$$X \mapsto \sum_{n=0}^{\infty} \frac{a_n}{n!} X^n$$

*We then have that, for the canonical inner product $(A_1, A_2) = \mathrm{tr}(A_1^{\mathsf{T}} A_2)$,*

$$(\mathrm{d}\phi)_X^* = (\mathrm{d}\phi)_{X^{\mathsf{T}}} \qquad X \in \mathbb{R}^{n \times n}.$$

*Proof.* We can compute the differential of $\phi$ as

$$(\mathrm{d}\phi)_X(E) = \sum_{n=0}^{\infty} \left( \frac{a_n}{n!} \sum_{i=0}^{n} X^i E X^{n-i} \right).$$

By linearity, it is enough to compute the adjoint of functions of the form $X \mapsto X^i E X^{n-i}$.

Observe that the adjoint of the left multiplication $L_A(X) = AX$ is exactly $L_{A^{\mathsf{T}}}$

$$\langle L_A(X), Y \rangle := \mathrm{tr}((AX)^{\mathsf{T}} Y) = \mathrm{tr}(X^{\mathsf{T}} A^{\mathsf{T}} Y) = \langle X, L_{A^{\mathsf{T}}}(Y) \rangle.$$

In the case of right multiplication, we also get $R_A^* = R_{A^{\mathsf{T}}}$.

Finally, we just have to apply this formula to the functions $L_{X^i}(E) = X^i E$ and $R_{X^{n-i}}(E) = E X^{n-i}$, and noting that $X \mapsto X^i E X^{n-i} = L_{X^i}(R_{X^{n-i}}(E))$, and that for any two functions, $(f \circ g)^* = g^* \circ f^*$, we get the result. $\qquad\square$

After obtaining this more general result, we thought that this should be folklore in some areas of functional analysis and numerical analysis. In fact, this result can be found without proof in Higham [2008, p.66].

**Remark.** The generalization of this result to complex functions is direct, just computing the differential of the analytic function with conjugate coefficients in its Taylor series. In this case, one can interpret this theorem by saying that "the adjoint of the differential is the differential of the conjugate at the adjoint", noting the two different meanings of the word *adjoint* in the sentence.

In the complex setting, one can formulate the theorem for a holomorphic function defined just on an open subset $U \subseteq \mathbb{C}$, and define the function on matrices on the set of matrices such that their spectrum is contained in $U$, hence making sense also of functions like $\log(X)$.

The result still holds true for many other inner product in $\mathbb{C}^{n \times n}$ (or $\mathbb{R}^{n \times n}$), in particular, for those for which for every matrix $X$ there exists a matrix $Y$ such that $L_X^* = L_Y$. If this is the case, we write $X^* := Y$ and the theorem still holds true, as in this case, $R_X^* = R_{X^*}$. Most of the scalar products on matrix spaces that appear in differential geometry have this property. For example, if we have a symmetric positive definite matrix $G \in \mathbb{R}^{n \times n}$ and we define the following product $\langle X, Y \rangle := \operatorname{tr}(X^\mathsf{T} G Y)$, then $X^* = (GXG^{-1})^\mathsf{T}$.

We can now state the case of $\exp(X)$ as a corollary of Theorem D.1 and Corollary B.2.

**Corollary D.2 (Gradient of the matrix parametrization).** *Let $f \colon \operatorname{GL}(n) \to \mathbb{R}$ be a smooth function, the gradient of $f \circ \exp$ at a matrix $A \in \mathfrak{gl}(n) \cong \mathbb{R}^{n \times n}$ with respect to the canonical metric at a matrix $B \in \operatorname{GL}(n)$, $\langle A_1, A_2 \rangle_B = \operatorname{tr}(A_1^\mathsf{T} A_2)$ is given by*

$$\nabla(f \circ \exp)(A) = (\mathrm{d}\exp)_{A^\mathsf{T}}(\nabla f(e^A)).$$

Using the chain rule, we can also compute the gradient with respect to the dynamic Lie trivialization $\exp_B$.

**Corollary D.3.** *Let $f \colon \operatorname{GL}(n) \to \mathbb{R}$ be a smooth function, and let $B \in \operatorname{GL}(n)$. The gradient of $f \circ \exp_B$ at a matrix $A \in T_B \operatorname{GL}(n) \cong \mathfrak{gl}(n) \cong \mathbb{R}^{n \times n}$ with respect to the canonical metric $\langle A_1, A_2 \rangle_B = \operatorname{tr}(A_1^\mathsf{T} A_2)$ is given by*

$$\nabla(f \circ \exp_B)(A) = (B^{-1})^\mathsf{T}(\mathrm{d}\exp)_{(B^{-1}A)^\mathsf{T}}(B^\mathsf{T} \nabla f(\exp_B(A))).$$

**Remark.** These two corollaries still hold if we replace $\operatorname{GL}(n)$ by any real matrix Lie group with this metric. The complex case is analogous.

**Remark.** In Lezcano-Casado and Martínez-Rubio [2019] the following slightly different formula for the gradient of the exponential is derived for compact real matrix Lie groups:

$$\nabla(f \circ \exp)(A) = e^A(\mathrm{d}\exp)_{-A}(e^{-A}\nabla f(\exp(A))).$$

This formula agrees with the one presented here, as it turns out that multiplication by $e^A$ commutes with $(\mathrm{d}\exp)_{-A}$. This can be seen, for example, modifying the proof of formula for the derivative of exponential map in Rossmann [2006, Chapter 1, Theorem 5] to obtain

$$(\mathrm{d}\exp)_A(X) = \sum_{k=0}^{\infty} \frac{(-\operatorname{ad}_A)^k}{(k+1)!}(e^A X).$$

Finally, if $G$ is a real compact matrix Lie group together with a bi-invariant metric, one has that for every $A \in \mathfrak{g}$, $A^* = -A$, where $A^*$ should be understood in the sense of $L_A^* = L_{A^*}$. This can be seen, for example, considering that a real compact matrix Lie group is either a subgroup of the orthogonal group or a conjugate of one. Using this, we finally see that the formula presented in Lezcano-Casado and Martínez-Rubio [2019] is equivalent to Corollary D.2.

# E   Examples of Matrix Manifolds and Specific Trivializations

This section has an expository purpose. It is intended as a compilation of useful results for the implementation of different trivializations. We will go over the forms that the Lie exponential and the Riemannian exponential—geodesics—take in different manifolds that are useful in the field of machine learning.

We will deliberately develop as least theory as possible, but we will still point out the relevant literature sources as *remarks*, for those interested in the theoretical background. At the end, we will also describe some retractions, which are useful for problems on which computing the geodesics or the Lie exponential is too expensive.

We will put as examples some Lie groups, the sphere and the hyperbolic space, the Stiefel manifold, and the space of symmetric positive definite matrices.

**Remark.** On some of the manifolds considered below, the metric is not the canonical one given by $\langle A_1, A_2 \rangle_B = \mathrm{tr}(A_1^\mathsf{T} A_2)$, but often a left-translation of this one of the form

$$\langle A_1, A_2 \rangle_B = \mathrm{tr}((B^{-1}A_1)^\mathsf{T} B^{-1} A_2) \qquad \forall A_1, A_2 \in T_B \mathcal{M}.$$

For these metrics, when we compute the gradient, we cannot use Corollary D.2 directly. On the other hand, after a similar reasoning, we get that the differential with respect to these metrics is given by the formula

$$\nabla(f \circ \exp)(A) = B(\mathrm{d}\exp)_{A^\mathsf{T}}(B^{-1}\nabla f(e^A)).$$

We can also deduce this formula just noting that, for these metrics, left translations are isometries by construction.

## E.1 Compact matrix Lie groups

On a Lie group, we can identify all the tangent spaces using left multiplication. In particular, we have tangent spaces at an arbitrary point can be identified with tangent space at the identity $\mathfrak{g} := T_e G$—the Lie algebra of $G$. The identification is given by

$$T_B G = \{BA \mid A \in \mathfrak{g}\} \qquad B \in G.$$

As such, if we know the structure of $\mathfrak{g}$ we can parametrize any tangent space of $G$.

For compact Lie groups, the Lie exponential and the Riemannian exponential agree[9] and take the form

$$\exp_{G,B}(\tilde{A}) = \exp_B(\tilde{A}) = B\exp(B^{-1}\tilde{A}) = B\exp(A) \qquad A \in \mathfrak{g}. \tag{2}$$

where $\exp$ is the exponential of matrices and we still used the identification $\tilde{A} = BA$. For these groups, the Riemannian exponential is surjective.

These Lie groups were already presented in Lezcano-Casado and Martínez-Rubio [2019] in the context of optimization for neural networks. In that paper, this trivialization was only considered in the static case, namely $\exp\colon \mathfrak{g} \to G$.

The gradient of Equation (2) is given by Proposition 6.1.

Finally, for compact matrix Lie groups, in order to use this formula to implement the dynamic trivialization method, we are just missing the expression for the Lie algebra $\mathfrak{g} \subseteq \mathbb{R}^{n \times n}$ of the Lie group in which we are interested. We give a list of some of these below.

**Special orthogonal group**

$$\mathrm{SO}(n) = \{B \in \mathbb{R}^{n \times n} \mid B^\mathsf{T} B = \mathrm{I}, \det B = 1\} \qquad \mathfrak{so}(n) = \mathrm{Skew}(n) = \{A \in \mathbb{R}^{n \times n} \mid A^\mathsf{T} = -A\},$$

**Unitary group**

$$\mathrm{U}(n) = \{B \in \mathbb{C}^{n \times n} \mid B^* B = \mathrm{I}\} \qquad \mathfrak{u}(n) = \{A \in \mathbb{C}^{n \times n} \mid A^* = -A\},$$

**Special unitary group**

$$\mathrm{SU}(n) = \{B \in \mathbb{C}^{n \times n} \mid B^* B = \mathrm{I}, \det B = 1\} \qquad \mathfrak{su}(n) = \{A \in \mathbb{C}^{n \times n} \mid A^* = -A, \mathrm{tr}\, A = 0\}.$$

**Complex torus**

$$\mathbb{T}(n, \mathbb{C}) = \{B \in \mathrm{Diag}(n, \mathbb{C}) \mid |B_{ii}| = 1\} \qquad \mathfrak{t}(n, \mathbb{C}) = \{A \in \mathrm{Diag}(n, \mathbb{C}) \mid A_{ii} \in i\mathbb{R} \subseteq \mathbb{C}\},$$

**Remark.** We say that $\mathbb{T}(n, \mathbb{C})$ is a torus because it is a product of $n$ circles. This can easily be seen simply defining the circle as $S^1 = \{z \in \mathbb{C} \mid |z| = 1\}$, so that $\mathbb{T}(n, \mathbb{C}) \cong S^1 \times \cdots \times S^1$. In this case, the correspondence between the Lie algebra and the Lie group is given by the Euler formula.

**Real torus**    The real torus $\mathbb{T}(2n, \mathbb{R})$ consists of the $2n \times 2n$ block-diagonal matrices with blocks of the form

$$\begin{pmatrix} \cos(\theta) & -\sin(\theta) \\ \sin(\theta) & \cos(\theta) \end{pmatrix} \qquad \theta \in [-\pi, \pi].$$

Note that is exactly the matrix representation of the complex number $e^{\theta i}$. In this case, its Lie algebra is given by the block-diagonal matrices with blocks given by

$$\begin{pmatrix} 0 & -a \\ a & 0 \end{pmatrix} \qquad a \in \mathbb{R}.$$

**Remark.** In the case of the real and complex torus, the exponential is a Riemannian covering map, this meaning that, in particular, it is always a local isometry, and it does not create local minima or saddle points. For this reason, to optimize on these two manifolds, we would not need to use dynamic trivializations, given that a static trivialization would work just fine as a direct corollary of Theorem 4.3.

## E.2    The groups $\mathrm{GL}^+(n)$ and $\mathrm{SL}(n)$

We then look at two more Lie groups on which we can compute the Riemannian exponential. These groups are of primal importance for problems that require invertible matrices or the study of volume-flows, like normalizing flows.

These groups are also an important example of groups on which the Riemannian exponential and the Lie exponential do not agree, and thus, in this case, we have two different trivialization schemes.

Furthermore, the Lie exponential is not surjective on these groups, so these are also examples of a retraction that could not be used as a static trivialization, but it can be used as a dynamic one.

**Positive general linear group**

$$\mathrm{GL}^+(n) = \{B \in \mathbb{R}^{n \times n} \mid \det B > 0\} \qquad \mathfrak{gl}(n) = \mathbb{R}^{n \times n}.$$

This is the connected component containing the identity matrix of the general linear group

$$\mathrm{GL}(n) = \{B \in \mathbb{R}^{n \times n} \mid \det B \neq 0\}.$$

**Special linear group**

$$\mathrm{SL}(n) = \{B \in \mathbb{R}^{n \times n} \mid \det B = 1\} \qquad \mathfrak{sl}(n) = \{A \in \mathbb{R}^{n \times n} \mid \operatorname{tr} A = 0\}.$$

**Remark.** The orthogonal projection from $\mathbb{R}^{n \times n}$ onto $\mathfrak{sl}(n)$ is given by

$$\pi_{\mathfrak{sl}(n)} \colon \mathbb{R}^{n \times n} \to \mathfrak{sl}(n)$$
$$A \mapsto A - \tfrac{1}{n} \operatorname{tr}(A)\mathrm{I}$$

We can use this formula to parametrize $\mathfrak{sl}(n)$, in the same way that we use $A \mapsto \frac{1}{2}(A - A^\intercal)$ to parametrize $\mathfrak{so}(n) \cong \mathrm{Skew}(n)$.

On these groups have two different trivializations based on the exponential of matrices.

On the one hand, we still have the dynamic Lie trivialization $\exp_B$ presented in Appendix E.1.

On the other hand, if $G$ is $\mathrm{GL}(n)$ or $\mathrm{SL}(n)$ for $n > 2$ equipped with the metric $\langle \tilde{A}_1, \tilde{A}_2 \rangle_B = \operatorname{tr}((B^{-1}\tilde{A}_1)^\intercal B^{-1}\tilde{A}_2)$, we have that the Riemannian trivialization for these groups is given by [10]

$$\exp_{G,B}(BA) = B \exp(A^\intercal) \exp(A - A^\intercal) \quad \text{for } A \in \mathfrak{g}.$$

Note that $BA \in T_B G$, as one would expect.

**Remark.** This result was first stated in Wang et al. [1969], and a proof of it can be found in Helgason [1979, Chapter 6, Exercise A.9]. For the proof for $\mathrm{GL}(n)$, see Andruchow et al. [2014, Theorem 2.14].

We can then compute the gradient of this parametrization as we know how to compute the gradient of the exponential map with respect to this metric, as detailed at the beginning of Appendix E.

**Remark.** It happens that the Lie exponential is not surjective on $\mathrm{SL}(n)$ so, in this case, it would not be possible to set $K = \infty$ in the dynamic trivialization algorithm, that is, it would be necessary to change the basis of the trivialization. The Lie trivialization is not surjective on $\mathrm{GL}^+(n, \mathbb{R})$ either, but it is surjective on $\mathrm{GL}(n, \mathbb{C})$, with $\mathfrak{gl}(n, \mathbb{C}) \cong \mathbb{C}^{n \times n}$.

These are examples for which using dynamic trivializations allow us to use certain parametrizations that we would not be able to use in the context of static trivializations.

The Riemannian exponential on $\mathrm{SL}(n)$ and $\mathrm{GL}(n, \mathbb{R})$ is surjective with this metric.

**Remark.** On these two manifolds, we can also use their polar decomposition as a trivialization to optimize over them, see Hall [2015, Proposition 2.19].

### E.3 Naturally reductive homogeneous spaces

In this section we touch on a few of the most used manifolds in optimization, namely the Stiefel manifold, the sphere, the hyperbolic space, and the symmetric positive definite matrices.

In this section we will restrict ourselves to expose the formulae for the exponential on these manifolds for certain metric. Most of these manifolds fall under the theory of symmetric manifolds, or the more general theory of naturally reductive homogeneous spaces. For a derivation of the fomulae in this section in the more general context of naturally reductive homogeneous spaces, we refer the reader to the self-contained exposition in Gallier and Quaintance [2019, Chapter 22].

#### E.3.1 Stiefel manifold

The Stiefel manifold is the manifold of $n \times k$ matrices with $k \leq n$ with orthonormal columns. Equivalently, it is the set of orthonormal $k$-frames on $\mathbb{R}^n$. In symbols we can see the Stiefel manifold as a submanifold of $\mathbb{R}^{n \times k}$ as follows:

$$\mathrm{St}(n, k) \coloneqq \{B \in \mathbb{R}^{n \times k} \mid B^\intercal B = \mathrm{I}_k\} \qquad T_B \mathrm{St}(n, k) = \{\tilde{A} \in \mathbb{R}^{n \times k} \mid B^\intercal \tilde{A} \in \mathfrak{so}(k)\}$$

Note that $\mathrm{St}(n, n) \cong \mathrm{O}(n)$. In this case, compare the formula of the tangent space with that given for $T_B \mathrm{SO}(n)$ Lie groups in Appendix E.1, in particular that of $\mathfrak{so}(n)$.

If we consider any completion of the frame $B$ into a basis of $\mathbb{R}^n$, that is, a matrix $B_\perp \in \mathbb{R}^{n \times n-k}$ such that $(B \; B_\perp) \in \mathrm{O}(n)$, we have the more computationally amenable description of the tangent spaces of $\mathrm{St}(n, k)$

$$T_B \mathrm{St}(n, k) = \{BA + B_\perp A_\perp \in \mathbb{R}^{n \times k} \mid A \in \mathfrak{so}(k), A_\perp \in \mathbb{R}^{n-k \times k}\}.$$

Note that if $n = k$, $T_B \mathrm{St}(n, n) = \{BA \mid A \in \mathfrak{so}(n)\}$ and we still recover the same definition from Appendix E.1.

The canonical metric [11] on the Stiefel manifold is given for $B \in \mathrm{St}(n, k)$, $\tilde{A}_1, \tilde{A}_2 \in T_B \mathrm{St}(n, k)$ by

$$\langle \tilde{A}_1, \tilde{A}_2 \rangle_B = \mathrm{tr}(\tilde{A}_1^\intercal (\mathrm{I}_n - \tfrac{1}{2} BB^\intercal) \tilde{A}_2)$$

With the notation as above, consider the QR decomposition $QR = (\mathrm{I}_n - BB^\intercal)\tilde{A}$ with $Q \in \mathrm{St}(n, k)$, $R \in \mathbb{R}^{k \times k}$, then the Riemannian exponential is given

$$\exp_{\mathrm{St}(n,k),B}(\tilde{A}) = (B \quad Q) \exp \begin{pmatrix} A & -R \\ R & 0 \end{pmatrix} \begin{pmatrix} I_k \\ 0 \end{pmatrix}.$$

**Remark.** The computational cost of computing geodesics on $\mathrm{St}(n, k)$ is then dominated by the computation of a thin-QR factorization of a $n \times k$ matrix and the computation of a exponential of a skew-symmetric $2k \times 2k$ matrix.

If $2k > n$, a more efficient algorithm is possible. We just have to compute the geodesics on $\mathrm{SO}(n)$ as per Appendix E.1 and then drop then project the result onto $\mathrm{St}(n,k)$ dropping the last $n-k$ columns. This process requires the computation of just one exponential of an $n \times n$ matrix. This process is equivalent to the formula described above.

**Remark.** In Edelman et al. [1998, Section 2.2.2] the authors give a formula for the geodesics of $\mathrm{St}(n,k)$ seen as a submanifold of $\mathbb{R}^{n \times k}$, that is, with the metric $\langle \tilde{A}_1, \tilde{A}_2 \rangle = \mathrm{tr}(\tilde{A}_1^\intercal \tilde{A}_2)$. In Section 2.4.1 they also discuss an essential difference between the Euclidean metric and the canonical metric on the Stiefel manifold.

### E.3.2    The sphere and the hyperbolic plane

The case of the sphere $S^n = \{x \in \mathbb{R}^{n+1} \mid \|x\| = 1\}$ is probably one of the most classical ones. We will always consider the *round sphere*, this is, the sphere as a subset of $\mathbb{R}^{n+1}$ together with the metric inherited from $\mathbb{R}^{n+1}$.

Its tangent space at a point $x \in S^n$ is simply given by the set of vectors orthogonal to it

$$T_x S^n = \{v \in \mathbb{R}^n \mid \langle x, v \rangle = 0\}.$$

and the geodesics are given by

$$\exp_{S^n, x}(v) = \cos(\|v\|)x + \sin(\|v\|)\frac{v}{\|v\|}.$$

To describe the $n$-dimensional hyperbolic space, first consider the diagonal matrix $\mathrm{I}_{n,1}$ with $n$ positive ones and a negative one on its diagonal. We will use the following notation

$$\langle x, y \rangle_{\mathbb{H}} := \langle x, \mathrm{I}_{n,1} y \rangle = \sum_{i=1}^n x_i y_i - x_{n+1} y_{n+1} \qquad \forall x, y \in \mathbb{R}^{n+1}$$

and denote by $\|x\|_{\mathbb{H}} = \sqrt{\langle x, x \rangle_{\mathbb{H}}}$ whenever $\langle x, x \rangle_{\mathbb{H}} \geq 0$.

With this notation, the $n$-dimensional hyperbolic space $\mathbb{H}^n$ can be seen as the submanifold of $\mathbb{R}^{n+1}$ defined by

$$\mathbb{H}^n = \{x \in \mathbb{R}^{n+1} \mid \langle x, x \rangle_{\mathbb{H}} = -1, x_{n+1} > 0\}$$

with tangent space at $x \in \mathbb{H}^n$ given by

$$T_x \mathbb{H}^n = \{v \in \mathbb{R}^{n+1} \mid \langle x, v \rangle_{\mathbb{H}} = 0\}.$$

The geodesics are then given by

$$\exp_{\mathbb{H}^n, x}(v) = \cosh(\|v\|_{\mathbb{H}})x + \sinh(\|v\|_{\mathbb{H}})\frac{v}{\|v\|_{\mathbb{H}}}.$$

**Remark.** The formula for the sphere is just a particular case of the one given for $\mathrm{St}(n+1, 1) \cong S^n$.

The reason why the formulas of the geodesics on the sphere and the hyperbolic plane are so similar has a geometric meaning. This can be seen in a more general case, considering the oriented Grassmannian manifold and the hyperbolic Grassmannian. The sphere and the hyperbolic plane are special cases of these manifolds. These manifolds are *symmetric spaces* and they are dual to each other. For more on the duality between symmetric spaces of compact and non-compact type, we refer the reader to Helgason [1979, Chapter 5, Example 1] or O'Neill [1983, Chapter 11].

**Remark.** In the same spirit as we can compute the geodesics on $\mathrm{St}(n,k)$ by taking a geodesic in $\mathrm{SO}(n)$ and projecting it down to $\mathrm{St}(n,k)$, we can also compute the geodesics of the real projective plane $\mathbb{RP}^n$ by computing the geodesic on $S^n$ and projecting it down to $\mathbb{RP}^n$. The metric induced on $\mathbb{RP}^n$ is called the *standard round metric* on $\mathbb{RP}^n$. If we perform the same process between $S^{2n+1}$ and $\mathbb{CP}^n$ and, in this case, we would get the *Fubini-Study* metric. These constructions are particular cases of the more general theory of Riemannian submersions. In particular, see O'Neill [1983, Chapter 7, Definition 44] for an introduction by one of the fathers of the theory, and Besse [2008, Chapter 9] for a more advanced review of the subject.

### E.3.3 The symmetric positive definite matrices

The symmetric positive definite matrices $\mathrm{Sym}^+(n)$ do not form a Lie group, as they are not closed under matrix multiplication, but they are a symmetric space.

When seen as a subset of $\mathbb{R}^{n \times n}$, we can endow it with a left-invariant metric defined as $\langle \tilde{A}_1, \tilde{A}_2 \rangle_B = \mathrm{tr}(B^{-1} A_1 B^{-1} A_2)$. The tangent space at a point $B \in \mathrm{Sym}^+(n)$ is given by

$$T_B \mathrm{Sym}^+(n) = \{ B^{1/2} A B^{1/2} \mid A \in \mathfrak{sym}(n) \}$$

where $\mathfrak{sym}(n)$ is the tangent space at the identity, given by the symmetric matrices

$$\mathfrak{sym}(n) = \{ A \in \mathbb{R}^{n \times n} \mid A^\mathsf{T} = A \}.$$

Note that for a symmetric positive definite matrix the square root is well defined, as symmetric positive definite matrices are diagonalizable, and the square root is just the matrix whose eigenvalues are the (positive) square root of the eigenvalues of the initial matrix.

Following the notation for Lie groups, if we denote $\tilde{A} = B^{1/2} A B^{1/2}$, we have that

$$\exp_{\mathrm{Sym}^+(n), B}(\tilde{A}) = B^{1/2} \exp(B^{-1/2} \tilde{A} B^{-1/2}) B^{1/2} = B^{1/2} \exp(A) B^{1/2}.$$

**Remark.** In this case, this manifold also constitutes an example of a symmetric space since $\mathrm{Sym}^+(n) \cong \mathrm{GL}^+(n)/\mathrm{O}(n)$. The metric considered here is the natural one with respect to this structure. An introduction to the computational aspects of this manifold can be found in Bonnabel and Sepulchre [2009].

### E.4 Some retractions

For now we have just mentioned examples regarding either the Lie exponential or the Riemannian exponential, but the dynamic trivialization framework allows us to use any function that is a retraction. In order to make use of arbitrary retractions, we just have to be able to compute the gradient of the function when precomposed with them. We will do so for a few important examples in this section.

In the case of the two retractions mentioned in Section 6.3, the Cayley map and projectors, their derivatives are already implemented in the major deep-learning packages, like Pytorch or Tensorflow. The first one just requires an inverse (or, more efficiently and stable, the solution of a system of the form $AX = B$) and the second one just requires the derivatives with respect to the SVD decomposition.

The retraction induced by a projector can be easily implemented for most manifolds. For example, for the sphere takes just the form $x \mapsto \frac{x}{\|x\|}$, whose derivative can also be computed just using autodiff.

For the symmetric positive definite matrices, we have the retraction from the symmetric matrices into the positive semidefinite matrices given by $A \mapsto A^2$. This one is similar to the frequently used from the upper triangular matrices given by the Cholesky decomposition $L \mapsto LL^\mathsf{T}$. The former has the advantage that we have access to $A$ which is the square root of its image. This can be helpful, as sometimes the square root of the matrix is needed for some computations, as we have seen in Appendix E.3.3. The retraction given by the Cholesky decomposition has the advantage that, if the diagonal of the upper-triangular matrix $L$ is strictly positive, then $LL^\mathsf{T}$ will be positive definite. For this reason this retraction is often used to parametrize variance kernels in Bayesian statistics.

Another retraction for $\mathrm{Sym}^+(n)$ is given by the exponential of matrices $\exp\colon \mathfrak{sym}(n) \to \mathrm{Sym}^+(n)$ which is a diffeomorphism. As such, it provides a rather good, although expensive, option to parametrize this manifold.

For a much more in-depth treatment of retractions, we refer the reader to Absil et al. [2009].

## F Detailed Experiment Set-Up and Hyperparameters

We tried to reproduce as faithfully as possible the set-up from previous experiments, to achieve a fair comparison. The batch size for all the experiments is 128. We fixed the seed to be 5544 of both Numpy and Pytorch for reproducibility in the final runs.

The exact architecture to process for a sequence of inputs $x_t \in \mathbb{R}^d$ with a hidden size $p$ is given by the formula

$$h_{t+1} = \sigma(\exp(A)h_t + Tx_{t+1})$$

with $A \in \text{Skew}(p)$ and $T \in \mathbb{R}^{p \times d}$. $\sigma$ is the `modrelu` non-linearity introduced in Arjovsky et al. [2016].

The initialization *Henaff* refers to initializing the diagonal blocks of the skew-symmetric matrices with elements sampled from the uniform distribution $\mathcal{U}(-\pi, \pi)$ as detailed in Henaff et al. [2016]. The *Cayley* initialization refers to sampling the diagonal from a distribution $u \sim \mathcal{U}(0, \pi/2)$ and then computing $s = -\sqrt{\frac{1-\cos(u)}{1+\cos(u)}}$ as detailed in Helfrich et al. [2018].

As we mentioned in the experiments section, we did not include the copying experiment that was usually used in previous papers, given that, as it was demonstrated in Lezcano-Casado and Martínez-Rubio [2019], the exponential trivialization converges to the exact solution even when based at the identity. The same happens when used with the dynamic trivialization, so we do not think that this experiment adds anything to the results.

Table 3: Hyperparameters for DTRIV1.

| Dataset | Size | Optimizer | Learning Rate | Orthogonal optimizer | Orthogonal Learning Rate |
|---|---|---|---|---|---|
| MNIST | 170 | | $10^{-3}$ | | $10^{-4}$ |
| | 360 | | $10^{-3}$ | | $10^{-4}$ |
| | 512 | RMSPROP | $5 \cdot 10^{-4}$ | RMSPROP | $7 \cdot 10^{-5}$ |
| P-MNIST | 170 | | $7 \cdot 10^{-4}$ | | $2 \cdot 10^{-4}$ |
| | 360 | | $7 \cdot 10^{-4}$ | | $7 \cdot 10^{-5}$ |
| | 512 | | $5 \cdot 10^{-4}$ | | $5 \cdot 10^{-5}$ |
| TIMIT | 224 | | $10^{-3}$ | | $10^{-4}$ |
| | 322 | ADAM | $10^{-3}$ | RMSPROP | $10^{-4}$ |
| | 425 | | $10^{-3}$ | | $10^{-4}$ |

Table 4: Hyperparameters for DTRIV100.

| Dataset | Size | Optimizer | Learning Rate | Orthogonal optimizer | Orthogonal Learning Rate |
|---|---|---|---|---|---|
| MNIST | 170 | | $5 \cdot 10^{-4}$ | | $10^{-4}$ |
| | 360 | | $3 \cdot 10^{-4}$ | | $5 \cdot 10^{-5}$ |
| | 512 | RMSPROP | $5 \cdot 10^{-4}$ | RMSPROP | $10^{-4}$ |
| P-MNIST | 170 | | $7 \cdot 10^{-4}$ | | $10^{-4}$ |
| | 360 | | $5 \cdot 10^{-4}$ | | $7 \cdot 10^{-5}$ |
| | 512 | | $5 \cdot 10^{-4}$ | | $5 \cdot 10^{-5}$ |
| TIMIT | 224 | | $10^{-3}$ | | $2 \cdot 10^{-4}$ |
| | 322 | ADAM | $10^{-3}$ | RMSPROP | $2 \cdot 10^{-4}$ |
| | 425 | | $10^{-3}$ | | $10^{-4}$ |

Table 5: Hyperparameters for DTRIV∞.

| Dataset | Size | Optimizer | Learning Rate | Orthogonal optimizer | Orthogonal Learning Rate |
|---|---|---|---|---|---|
| MNIST | 170 | | $7 \cdot 10^{-4}$ | | $10^{-4}$ |
| | 360 | | $5 \cdot 10^{-4}$ | | $10^{-4}$ |
| | 512 | RMSPROP | $10^{-4}$ | RMSPROP | $7 \cdot 10^{-5}$ |
| | 170 | | $7 \cdot 10^{-4}$ | | $2 \cdot 10^{-4}$ |
| P-MNIST | 360 | | $7 \cdot 10^{-4}$ | | $5 \cdot 10^{-5}$ |
| | 512 | | $3 \cdot 10^{-4}$ | | $7 \cdot 10^{-5}$ |
| TIMIT | 224 | | $10^{-3}$ | | $2 \cdot 10^{-4}$ |
| | 322 | ADAM | $10^{-3}$ | RMSPROP | $2 \cdot 10^{-4}$ |
| | 425 | | $10^{-3}$ | | $2 \cdot 10^{-4}$ |

## Footnotes

[8]See the remark after the proof of the theorem.

[9]Here we are assuming that we consider the group $G$ together with a bi-invariant metric. For compact matrix Lie groups this metric is exactly $\langle A_1, A_2 \rangle_B = \mathrm{tr}(A_1^\mathsf{T} A_2)$. For more on this, we refer the reader to Lezcano-Casado and Martínez-Rubio [2019, Appendix C.1.].

[10]This result applies not only to $\mathrm{SL}(n)$, but to any semisimple Lie group equipped with the left-invariant metric associated to the Killing form.

[11]We say that this is the canonical metric because it is the one inherited from the quotient structure—as a homogeneous space—of $\mathrm{St}(n, k)$ as $\mathrm{St}(n, k) \cong \mathrm{SO}(n)/\mathrm{SO}(n-k)$. If we put the Euclidean metric $\frac{1}{2}\mathrm{tr}(X^\intercal Y)$ on $\mathrm{SO}(n)$, this metric is bi-invariant under the action of $\mathrm{SO}(n-k)$ and descends into the canonical metric on the quotient manifold $\mathrm{SO}(n)/\mathrm{SO}(n-k)$ described here. For the exact computations see Edelman et al. [1998].