[Reviews · NeurIPS 2019]

Reviewer 1



Lines 11-12 (and in other places): "the gradient of the exponential of matrices" -- the word "gradient" seems to be inadequate in this context, since gradients apply to functions that map to real numbers. From the appendix, it seems the authors here mean the gradient of a (real) function of the exponential of a matrix. Either way, in the end, the such gradients really only require the Fréchet derivative (or directional derivative) of the exponential map, and the adjoint of these differentials. This is indeed what the appendix derives, hence, perhaps the authors could reformulate in this sense? On a related note, Propostion 6.1 about the gradient of functions involving the matrix exponential: it is said after this proposition that the reference (Al-Mohy and Higham) gives an approximation of dexp. Perhaps it was a different paper, but my recollection is that this paper gives an exact formula, based on computing the exponential of the block matrix [X H ; 0 X] then extracting the top-right block, where X is the base point and H is the direction? This is mathematically exact, and gives machine precision in practice. Line 42, "... consequently not altering the convergence of the optimization method" -- This sentence suggests that changing the Riemannian metric on the manifold has no bearing on convergence, which is not true (think for example of Newton's method, which can be though of as running gradient descent with the Riemannian metric dictated by the Hessian of the cost function). The statement in Theorem 4.3 is mathematically imprecise, as the main claim is an "equivalence" that is not defined. Please rephrase. It seems that the actual claim is rather that critical points are in one-to-one correspondence through phi. I imagine the same is true for second-order critical points? Def. 3.1: the map r should be smooth for this to be a retraction. Line 138, "almost all the manifold": do you mean on almost the whole manifold? Or do you mean on almost any manifold? Either way, what is the meaning of "almost all" here? Line 161: "When working on a manifold, we do not have a notion of the Lebesgue measure of a set." -- Since we are here working on a Riemannian manifold, can't we have a notion of measure from the Riemannian metric? Question to the authors, out of curiosity: do you envision that it would be easy (and beneficial) to extend these techniques to second order optimization? It seems this would be impeded by the necessity of computing the second derivative of the matrix exponential, which may not be trivial computationally?

Reviewer 2



This paper studies a parametrization technique, which is called trivialization, for transforming a Riemannian manifold optimization problem to an equivalent problem in the Euclidean space. Algorithms used for Euclidean setting can then be used to solve this problem. Two useful trivializations are proposed: Riemannian trivialization and Lie trivialization. My main concern is the practicality of the proposed approach. It is not clear how to use the proposed trivializations to some common and simple manifolds: sphere and Stiefel manifold. Moreover, the computational cost of the trivialization seems to be still high, given that matrix exponential needs to be computed/approximated. Furthermore, what is the benefit of converting the manifold optimization problem back to Euclidean space? This ruins the possibly nice structure of the manifold, which may leads to gains on dimensionality reduction etc. ======== After reading the authors' rebuttal and other reviewers' reports, I am willing to increase my score, as some of my concerns are addressed appropriately.

Reviewer 3



This work provides a new way of optimization on manifold through combination of parameterization and Riemannian gradient descent. The paper is very well written and the organization is clear. The experimental section is convincing and the references are adequate. The methods and proofs are technically sound. The paper also made sufficient connections and contrasts with the existing methods. The code is included in the submission for reproduciblity check. Significance: The paper proposed a more general framework for optimization through parameterizations than the state-of-the-art method and showed improvement in several numerical experiments.

[Author Response · NeurIPS 2019]

We would like to thank all the reviewers for their effort, and their thoughtful comments. We are glad that the reviewers appreciated our contribution, and we will do our best to address the objections and minor errata that were pointed out.

**Rev1** L 11-12. Absolutely. Being formal, it should be "the gradient associated to the pullback of $f$ along $\exp$". We will change it to "the gradient with respect to the parametrization". Prop 6.1. This was a different paper by the same authors based on the paper cited here and published also in 2009. In the paper cited here, they just establish the state of the art for computing the exponential of matrices. For the gradient, we implement the method based on the exponential of the block matrix $[X, H; 0, X]$ for simplicity (*cf.*, line 374 in the file `exp_numpy.py`). The standard reference for this method is: Roy Mathias. *A chain rule for matrix functions and applications.* (1996). L 42. It is indeed misleading. We will change it to "on which standard convergence results still apply". Thm 4.3 We will change "is equivalent" to "accounts for". The same can be said about higher order methods. Def 3.1. Absolutely. L 138 and 161. Indeed, we have the volume form and the measure induced by the metric. We chose not to mention them in the main paper for simplicity. In l.138 we do mean "in almost all the manifold" in a measure-theoretical sense with respect to a measure induced by some metric. These two things indeed deserve a clarifying footnote. 2nd order. We believe that this could be extended to that framework. By the definition of the Hessian, one would need to be able to compute the covariant derivative of the adjoint of the exponential (or whichever retraction you are working with). This can be done for most manifolds on which you can compute geodesics (e.g. naturally reductive homogeneous spaces).

**Rev2** Regarding the efficiency concerns, we would like to note that, although the main examples that we presented were based on the exponential maps, one of the main contributions of the paper is to extend the framework in (Lezcano-Casado 2019) to retractions through trivializations and dynamic trivializations. Retractions are the way to perform cheap optimization on manifolds (*cf.*, Section 2). Also Note that, as pointed out in point 1. below, in the context of RNNs, the use of a retraction does not yield any time improvement over the exponential. On the gains of converting the problem to $\mathbb{R}^n$, one is able to use well-understood optimization methods developed for $\mathbb{R}^n$ that do not have generalizations for manifolds. **1.** The DTRIV methods come at no extra cost compared to regular trivializations. When compared to other methods like computing the Cayley map, computing the exponential of matrices is just twice as slow. Now, we use an implementation trick, by which using these and other parametrizations have a computationally negligible cost (*cf.*, Rev3, point 2). The final cost in CPU time of computing the exponential or the Cayley map is $\mathcal{O}(n^3)$ per iteration, where $n$ is the hidden size (same as a naïve multiplication of matrices), vs. $\mathcal{O}(b\ell n^2)$ of computing backprop, where $b$ is the batch size, and $\ell$ is the average length of the processed sequences. Furthermore, note that the Cayley map is also a retraction, so it also can benefit from being implemented as a DTRIV. We will add this point and examples of other retractions to section E. **2.** We chose RMSPROP for most experiments because it was the optimizer used in the other papers, and we wanted to show a fair comparison with the other methods. It might be possible to get better results than the ones shown in the paper with other optimizers, but we believe that this is very much problem-specific, so we preferred to stick to what we believe would be the fairest comparison. **3.** The case of the sphere and the Stiefel manifold can easily be solved by looking at them in the context of reductive homogeneous spaces, and deduce the formula of their geodesics from this. A standard reference for this is P.A Absil *Optimization Algorithms on Matrix Manifolds*. The geodesics in this case can be expressed in terms of the exponential of matrices, and since we proved in Prop 6.1 a formula to compute the gradient with respect to the exponential of matrices, we can then implement a DTRIV version of them in these manifolds. We will include the sphere and the Stiefel manifold examples worked out, as well as the hyperbolic space (useful for word embeddings) and how to deal with some standard retractions like the Cayley map or the QR retraction on the Stiefel manifold, or following a Euclidean geodesic and projecting back in the sphere. We hope this makes the paper more accessible to non-experts.

**Rev3 1.** The Riemannian and the Lie parametrization on compact Lie groups agree, so in this case it would be the same. For other groups (or homogeneous spaces) the methods do not generally agree, (*cf.*, section E). In any case, both of them can be shown to converge in the dynamic trivialization setting, as per Thm 4.3. and the discussion in sec 4.1 and 4.2. With a bit more work, one can show rates of convergence on matrix Lie groups for Lipschitz functions, matching with exactly the same constants, those of Riemannian gradient descent, but thit is outside of the scope of this paper. **2.** The exponential and its gradient takes about twice the time to approximate than the Cayley and its gradient. Now, we implement the trick outlined in Section 4.3 in (Lezcano-Casado 2019). Using this trick, the computation of the parametrization is negligible both for the exponential and the Cayley, compared to the cost of computing the whole backpropagation step. See the section in the paper mentioned above for an in-depth discussion. Moreover, we note that the Cayley map is also amenable to use in the DTRIV context, and it enjoys the same favorable properties compared to just using the naïve Cayley approach. As mentioned in Rev2, 1), we will include these and other examples in the final version of the paper. **3.** On the theoretical side, we have not pursued a detailed analysis, but this can be carried using ideas similar to those in DW Dreisigmeyet *Direct Search Methods on Reductive Homogeneous Spaces* (2018). From the practical point of view, we observed that choosing $K = \infty$ was usually good enough for most practical purposes, and we will suggest to do so in the paper. We leave for future research to benchmark the empirical performance of dynamical schedules for $K$.

[Meta-Review · NeurIPS 2019]

The reviews are overall positive, though not over the board. This work can be strengthened further by investigating further ML applications and settings.